# Nonalcoholic Fatty Liver Disease in Patients with Inherited and Sporadic Motor Neuron Degeneration

**DOI:** 10.3390/genes13060936

**Published:** 2022-05-24

**Authors:** Brian Johnson, Angela Kokkinis, Neville Gai, Ejaz A. Shamim, Craig Blackstone, Kenneth H. Fischbeck, Christopher Grunseich

**Affiliations:** 1Neurogenetics Branch, National Institute of Neurological Disorders and Stroke, NIH, Bethesda, MD 20892, USA; johnsonbrj@nih.gov (B.J.); akokkinis@cc.nih.gov (A.K.); cblackstone@mgh.harvard.edu (C.B.); fischbeck@ninds.nih.gov (K.H.F.); 2Radiology and Imaging Sciences, Clinical Center, NIH, Bethesda, MD 20892, USA; neville.gai@nih.gov; 3Mid-Atlantic Permanente Research Institute, Rockville, MD 20852, USA; ejaz.a.shamim@kp.org

**Keywords:** steatosis, motor neuron disease, amyotrophic lateral sclerosis, fatty liver

## Abstract

We describe evidence of fatty liver disease in patients with forms of motor neuron degeneration with both genetic and sporadic etiology compared to controls. A group of 13 patients with motor neuron disease underwent liver imaging and laboratory analysis. The cohort included five patients with hereditary spastic paraplegia, four with sporadic amyotrophic lateral sclerosis (ALS), three with familial ALS, and one with primary lateral sclerosis. A genetic mutation was reported in nine of the thirteen motor neuron disease (MND) patients. Fatty liver disease was detected in 10 of 13 (77%) MND patients via magnetic resonance spectroscopy, with an average dome intrahepatic triacylglycerol content of 17% (range 2–63%, reference ≤5.5%). Liver ultrasound demonstrated evidence of fatty liver disease in 6 of the 13 (46%) patients, and serum liver function testing revealed significantly elevated alanine aminotransferase levels in MND patients compared to age-matched controls. Fatty liver disease may represent a non-neuronal clinical component of various forms of MND.

## 1. Introduction

Motor neuron diseases (MNDs) are widely heterogeneous due to both the phenotypic diversity of the affected cell types and variation in underlying molecular defects [1]. Studies of MND patients have increasingly revealed evidence of non-neuronal involvement [2,3]. To comprehensively address the management of MND patients, a better understanding of non-neuronal MND manifestations is needed in a cohort with diverse features of MND with both genetic and sporadic etiologies.

Non-alcoholic fatty liver disease (NAFLD) is the most common liver disease in the world, with a global prevalence of nearly 25% [4]. The hallmark of the disease is hepatic steatosis with the accumulation of excess fat in the liver and metabolic dysfunction. Abnormalities in fat accumulation and fatty acid metabolism have previously been identified in patients with MND affecting both upper (UMNs) and lower motor neurons (LMNs), such as amyotrophic lateral sclerosis (ALS) [2], as well as spinal muscular atrophy (SMA) from deletion in the *SMN1* gene, which selectively targets LMNs [3]. Recently, we reported an increased incidence of NAFLD in patients with inherited LMN involvement from spinal and bulbar muscular atrophy (SBMA) [5]. The aim of the current study was to investigate the prevalence and features of NAFLD in a broader cohort of patients with MND.

## 2. Materials and Methods

Patients were recruited under the NIH Combined Neuroscience Institutional Review Board protocol NCT02124057, and informed written consent was obtained. Participants were screened by a neurologist and were required to have a diagnosis of MND other than SBMA. We enrolled male patients with MND who were age-matched (±5 years) to SBMA male patients and male controls from a previous study [5]. Fasting blood was collected and analyzed at the NIH Clinical Center Department of Laboratory Medicine. Liver fat content was measured by means of ^1^H magnetic resonance spectroscopy, and triacylglycerol was quantified at the dome of the right hepatic lobe, right inferior lobe, and left lobe as previously described [5].

## 3. Results

Thirteen male patients (mean age: 56 ± 13 years) with four different types of MND (five hereditary spastic paraplegia (HSP), four sporadic ALS, three familial ALS, one primary lateral sclerosis (PLS)) underwent clinical evaluation, including liver imaging and laboratory testing. A genetic etiology of disease was reported in nine of the thirteen MND patients. Ten patients in the MND cohort (77%) had evidence of NAFLD according to one or more imaging modality. Magnetic resonance spectroscopy (MRS) showed elevated (>5.5%) liver dome intrahepatic triacylglycerol (IHTG) in 8 of 11 (73%) subjects receiving MRI, with an average liver dome IHTG of 17.3% (SD: 18.4, range 1.8–63.1, Table 1). Evidence of elevated IHTG was also detected in the left and right lobes of the liver.

Features of NAFLD were observed in each of the four types of MND evaluated and in seven of the nine subjects with inherited forms of disease. Liver ultrasound showed evidence of hepatic steatosis in 46% (6/13) of MND patients, with four assessed as moderate and two assessed as mild. The MND cohort showed increased liver dome IHTG compared to our previously published cohort of 14 male controls (*p* = 0.0086, Figure 1A) [5].

Upon laboratory evaluation, ALT was significantly greater in the MND cohort compared to male controls, with seven subjects having elevated (>30 U/L) values (mean MND ALT: 31.9 ± 11.2 U/L, control ALT: 20.6 ± 6.3 U/L, *p* = 0.0031). Elevation of the Homeostatic Model Assessment for Insulin Resistance (HOMA-IR) was detected in eight (62%) of the MND cohort, seven of whom had liver imaging consistent with NAFLD. MND patient serum γ-glutamyl transferase (GGT) levels positively correlated with both liver dome IHTG (*r* = 0.58, *p* = 0.0063, Figure 2A) and ALT (*r* = 0.41, *p* = 0.018, Figure 2B), although no difference in GGT levels was observed between the MND and control cohorts. Other laboratory measures, including serum creatine kinase (CK), total bilirubin, and free fatty acids, were normal (Table 1).

## 4. Discussion

We report evidence of NAFLD in a cohort of MND patients, indicating that NAFLD may be a part of the clinical spectrum in sporadic and inherited forms of MND. We observed liver dome IHTG elevation in MND patients that was comparable to levels reported previously in SBMA [5]. We observed elevated ALT and evidence of insulin resistance in MND patients compared to controls. Interestingly, both liver dome IHTG and serum ALT were positively correlated with serum GGT in the MND cohort. Previous studies have reported associations between GGT levels and the onset of neurodegenerative diseases such as Parkinson’s disease [6] and dementia [7]. It is unclear if fatty deposition precedes GGT elevation in MND.

The presentation of NAFLD in both MND and SBMA may be a consequence of alterations in lipid metabolism in these diseases. Most of the MND subjects in our study were found to have evidence of both insulin resistance and NAFLD, indicating a potential link between these processes. In a previous study, evidence of hepatic steatosis was reported in 76% of ALS patients and only 19% of non-MND neurological disease cases, which included patients with movement disorders and dementia, suggesting that the metabolic findings may be unique to those with motor neuron degeneration [2]. Additionally, our previous study of NAFLD in SBMA showed distinct hepatic gene expression changes in SBMA patient liver samples compared to others with non-alcoholic steatohepatitis [5].

Several therapeutic strategies have been identified for patients with NAFLD. Emerging evidence suggests that anti-diabetic drugs can reduce fatty accumulation and decrease liver enzyme levels in NAFLD [8]. Studies have shown the benefit of antioxidants on fatty liver progression [9], and antioxidants such as vitamin E [10,11] have also been investigated as potential therapies for ALS.

Since the present study was a cross-sectional analysis of NAFLD in MND, we are unable to report on the age of NAFLD onset in this cohort. NAFLD onset is challenging to detect as it typically results in few or no symptoms [4]. One potential mechanism of NAFLD in our cohort may be the reduction in activity levels associated with muscle atrophy and functional deficits. Cross-sectional studies have highlighted a strong positive association between sedentary time and NAFLD [12,13], suggesting that MND patients may develop NAFLD due to reduced overall mobility. Other evidence from animal models of MND, such as mouse models of SMA from SMN1 deletion, have suggested that the NAFLD phenotype may precede findings of motor dysfunction [3]. Further research is necessary to understand the prevalence of NAFLD in MND and to characterize the relationship of this process with the underlying disease mechanisms.

## Figures and Tables

**Figure 1 genes-13-00936-f001:**
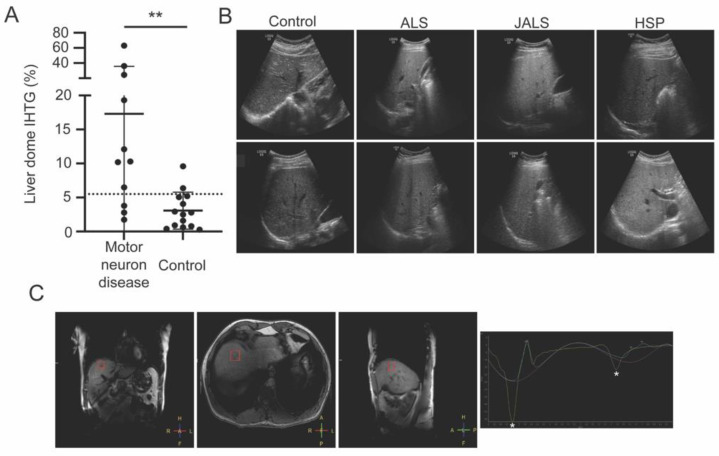
(**A**) ^1^H magnetic resonance spectroscopic imaging showed that the motor neuron disease group had a significantly higher intrahepatic triacylglycerol (IHTG) content in the dome of the liver compared to controls (** *p* = 0.0086); (**B**) increased echogenicity on liver ultrasound showing steatosis in patients with amyotrophic lateral sclerosis (ALS), juvenile ALS (JALS) with mutation in senataxin, and hereditary spastic paraplegia (HSP). Two patients from each category are shown; (**C**) voxel placement (red square) in the liver dome of a patient with HSP and spectroscopy trace showing the peaks for water (left asterisk) and triacylglycerol (right asterisk).

**Figure 2 genes-13-00936-f002:**
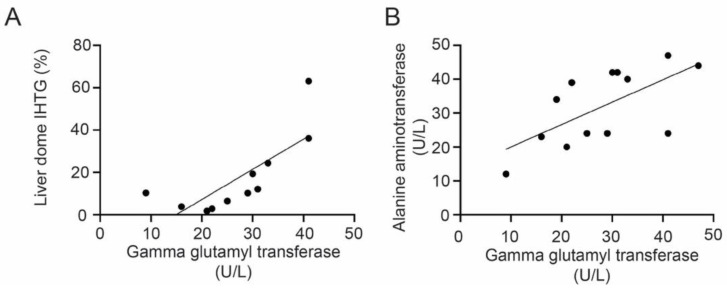
There was a significant correlation between (**A**) the liver dome intrahepatic triacylglycerol (IHTG) content and the serum γ-glutamyl transferase (*r* = 0.58, *p* = 0.0063) and (**B**) serum alanine aminotransferase and serum γ-glutamyl transferase (*r* = 0.41, *p* = 0.018) in the motor neuron disease group.

**Table 1 genes-13-00936-t001:** Motor neuron disease cohort characteristics.

Subject	Age/ Onset (years)	BMI (kg/m^2^)	Disease	Gene	Medications *	CK (U/L)	Total Bilirubin (mg/dL)	ALT (U/L)	HOMA-IR	GGT (U/L)	Cholesterol (mg/dL)	Triglyc-erides (mg/dL)	Free Fatty Acid (mEq/L)	Liver Dome IHTG (%)	Right Lobe IHTG (%)	Left Lobe IHTG (%)	Ultrasound
MND01	59/49	N/A	PLS	N/A	Tizanidine	109	0.5	24	N/A	25	187	80	0.5	6.5	8.2	4.3	Normal
MND02	68/13	N/A	ALS4	*SETX*	Atorvastatin	314	0.4	24	7.7	29	154	116	1.0	10.2	4.8	5.8	Normal
MND03	67/67	33	ALS	N/A	Atorvastatin, (AMX0035 research compound)	386	0.6	39	3.1	22	134	76	0.5	2.8	6.3	4.0	Normal
MND04	74/49	24	HSP	*SPG31*	Atorvastatin, solifenacin succinate	285	0.7	23	1.2	26	127	55	0.3	3.8	3.3	10.6	Normal
MND05	56/30	N/A	HSP	*SPG4*	Diltiazem	107	0.9	20	1.4	21	269	64	0.3	1.8	3.0	5.1	Normal
MND06	56/54	24	ALS	N/A	Riluzole	430	0.4	42	19.1	30	130	220	0.3	19.3	12.9	5.9	Moderate hepatic steatosis
MND07	29/19	N/A	HSP	*KIF1A*	None	148	0.6	47	4.6	41	197	197	0.8	63.1	8.6	10.3	Minimal hepatic steatosis
MND08	40/13	26	ALS4	*SETX*	None	200	0.5	40	2.8	33	137	80	0.3	24.4	17.6	3.6	Moderate hepatic steatosis
MND09	54/50	27	HSP	*SPG4*	None	189	1.6	24	2.7	41	229	133	0.5	36.1	9.3	38.4	Mild hepatic steatosis
MND10	67/30	21	HSP	*SPG4*	Duloxetine	250	0.5	12	1.5	9	155	55	0.5	10.3	0.0	3.3	Normal
MND11	59/58	24	ALS	N/A	Riluzole	1275	0.9	42	0.9	31	217	79	0.5	12.1	7.6	8.7	Normal
MND12	37/13	33	ALS4	*SETX*	None	177	0.5	44	4.9	47	175	100	0.4	^#^ Diffuse hepatic steatosis	^#^ Diffuse hepatic steatosis	^#^ Diffuse hepatic steatosis	Moderate hepatic steatosis
MND13	57/54	N/A	ALS	*C9orf72*	None	234	0.4	34	4.0	19	204	86	0.8	N/A	N/A	N/A	Moderate hepatic steatosis

* Medications listed are those which meet the criteria for the highest drug-induced liver injury (DILI) concern. Subjects MND06 and MND11 were on riluzole. ^#^ Spectroscopy not available and determination of hepatic steatosis made based on MRI imaging. Abbreviations: amyotrophic lateral sclerosis (ALS), alanine aminotransferase (ALT), body mass index (BMI), creatine kinase (CK), γ-glutamyl transferase (GGT), homeostatic model assessment insulin resistance index (HOMA-IR), hereditary spastic paraplegia (HSP), intrahepatic triacylglycerol (IHTG), kinesin family member 1A (KIF1A), primary lateral sclerosis (PLS), senataxin (SETX), spastic paraplegia 4 (SPG4), spastic paraplegia 31 (SPG31). N/A: Not Applicable.

## Data Availability

All data generated or analyzed during this study are included in this published article.

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
