# Peer review of "Nonalcoholic Fatty Liver Disease in Patients with Inherited and Sporadic Motor Neuron Degeneration"

_genes, 2022, doi:10.3390/genes13060936_

Round 1
Reviewer 1 Report
This manuscript determines the prevalence of non-alcoholic fatty liver disease in a specific cohort of patients with motor neuron disease. The manuscript is well written and highlighted the importance of altered lipid or fat metabolism in such neurodegenerative disorders. This study further supports in a clinical perspective with early studies carried using different model systems.
Author Response
We thank the reviewer for their support of our manuscript.
Reviewer 2 Report
The short comunication titled “Nonalcoholic fatty liver disease in patients with inherited and sporadic motor neuron degeneration” deals with an interesting topic.
The contents of the work are good and adequate.
However I have small suggestions:
First of all it is necessary to describe at what age the patients manifested the disease, if there are therapies in place and above all if they are men or women or if it is a mixed population.
Also, if there are any experimental therapies in place and if they will be practiced in the future.
Also the biochemistry of these patients ??????
I know it is a short communication but these small data are necessary to validate the work done and to support the continuation.
Another thing to highlight is the role of the liver enzyme GGT which plays a key role in hepatic fibrosis and liver disease and therefore the connection between its expression and the onset of neurodegeneration (doi: 10.1155 / 2018/5045734, doi: 10.1074 / jbc.RA119.009304; https://doi.org/10.3389/fnagi.2021.630409; https://doi.org/10.1038/s41598-020-58306-x)
I recommend making these changes to improve communication and the data shown
Author Response
We thank the reviewer for their review of our work and feel that the changes have helped to improve the quality of the manuscript. We have addressed the concerns as follows.
The short comunication titled “Nonalcoholic fatty liver disease in patients with inherited and sporadic motor neuron degeneration” deals with an interesting topic.
The contents of the work are good and adequate.
We thank the reviewer for their support of our manuscript.
However I have small suggestions:
First of all it is necessary to describe at what age the patients manifested the disease, if there are therapies in place and above all if they are men or women or if it is a mixed population.
We have added the age of MND disease onset to Table 1.
The MND patients enrolled in this study were males both age and sex matched to our previous cohort of patients with spinal and bulbar muscular atrophy (SBMA) and healthy control male cohorts. This information is included in the materials and methods section on line 51. SBMA is an X-linked disease that only affects men. We have specified in the first sentence of the results that only male MND patients were evaluated (line 58).
The age of NAFLD onset is not clear as this was a cross-sectional analysis of NAFLD in our MND cohort. We have added the following text to lines 199-201 of the discussion section.
“Since the present study was a cross-sectional analysis of NAFLD in MND, we are unable to report on the age of NAFLD onset in this cohort. NAFLD onset is challenging to detect as it typically results in few or no symptoms.”
Also, if there are any experimental therapies in place and if they will be practiced in the future.
We thank the reviewer for this suggestion. We have added information on therapies for NAFLD and ALS to the discussion section on lines 130-198.
“Several therapeutic strategies have been identified for patients with NAFLD. Emerging evidence suggests that anti-diabetic drugs can reduce fatty accumulation and decrease liver enzyme levels in NAFLD [8]. Studies have shown the benefit of antioxidants on fatty liver progression [9], and antioxidants such as Vitamin E [10,11] have also been investigated as potential therapies for ALS.”
Also the biochemistry of these patients ??????
I know it is a short communication but these small data are necessary to validate the work done and to support the continuation.
Another thing to highlight is the role of the liver enzyme GGT which plays a key role in hepatic fibrosis and liver disease and therefore the connection between its expression and the onset of neurodegeneration (doi: 10.1155 / 2018/5045734, doi: 10.1074 / jbc.RA119.009304; https://doi.org/10.3389/fnagi.2021.630409; https://doi.org/10.1038/s41598-020-58306-x)
We thank the reviewer for their recommendation to evaluate the biochemistry further. We found that there is a significant correlation between the liver dome intrahepatic triacylglycerol (IHTG) content and the serum gamma-glutamyl transferase (r = 0.58, p = 0.0063) and serum alanine aminotransferase and serum gamma-glutamyl transferase (r = 0.41, p = 0.018) in the motor neuron disease group. This information has been added to the results section in figure 2.
We have also added the following information to the discussion section at lines 115-120.
“We observed elevated ALT and evidence of insulin resistance in MND patients compared to controls. Interestingly, both liver dome IHTG and serum ALT were positively correlated with serum GGT in the MND cohort. Previous studies have reported associations between GGT levels and onset of neurodegenerative diseases such as Parkinson’s disease [6] and dementia [7]. It is unclear if fatty deposition precedes GGT elevation in MND.”